# OpenReview forum: "Removing Noise, not Finding Gold: Quality Filtering for Large-Scale Pretraining"
_ICML.cc/2026/Conference — ICML 2026 regular_

### Official Review · Reviewer_fo8q · 2026-02-18

**Soundness:** 3
**Presentation:** 3
**Significance:** 3
**Originality:** 2
**Overall Recommendation:** 5
**Confidence:** 3

**Summary:**

The paper provides an in-depth analysis of Classifier-based Quality Filtering (CQF) for large-scale pretraining. It shows that CQF can improve downstream performance without necessarily improving LM performance on the HQ set, and strikingly reports that training on CQF-selected data can outperform training directly on a sufficiently large HQ set. The authors explain this “paradox” by arguing CQF implicitly filters/reweights the HQ set itself, and they introduce an optimization-driven notion of data quality that can be estimated via small proxy experiments.

**Compliance With Llm Reviewing Policy:**

Affirmed.

**Final Justification:**

My concerns were resolved. I have therefore raised my score from 4 to 5.

**Key Questions For Authors:**

1. Can you better characterize conditions/failure modes where implicit HQ filtering aligns (or misaligns) with downstream tasks?
2. How robust are proxy-based quality estimates across model sizes and training recipes?
3. How do you detect/mitigate cases where CQF becomes overly specialized toward certain benchmarks?

**Limitations:**

yes

(Overall Recommendation: Could become a 5 with stronger reproducibility (code, configs, and scripts) and clearer key analyses.)

**Strengths And Weaknesses:**

Pros: Strong empirical/analytical narrative that challenges a common belief about CQF; compelling paradox + explanation via implicit HQ filtering.

Methodological care: Avoids confounding from too-small k causing token scarcity/repetition, focusing on quality effects.

Cons: Would benefit from a more structured taxonomy of when HQ alignment helps vs hurts, and stronger evidence that proxy quality estimates remain stable across scales/recipes.

Significance: High practical relevance given CQF’s role in major pretraining pipelines and public corpora.

Originality: Reinterpreting CQF as a distribution-shaping/optimization operator is a meaningful conceptual contribution.

---

> ### Author Rebuttal · Authors · 2026-03-30
>
> We thank the reviewer for their positive assessment and for recognizing the paper's narrative, methodological care, and practical significance. Below we address each concern and question.
>
> > **Taxonomy of when HQ alignment helps vs. hurts**
>
> We agree, and the ingredients already exist across our experiments. Appendix D (Figures 18-20) studies **how different HQ sets interact with downstream tasks**: KnowledgePile aligns most closely with ARC, while OpenOrca aligns better with MMLU (Figure 6). Appendix C (Figures 13-15) identifies failure modes where classifiers latch onto spurious features (e.g., sequence length bias with OpenOrca).
>
> We will consolidate these into a structured summary table organized by HQ set, downstream task, and selection fraction k. Thank you for this suggestion.
>
> > **Proxy quality estimates stability across scales/recipes**
>
> The CQF classifier is trained once (sBert + logistic regression on a fixed HQ set) and reused for models of all sizes. Figure 11 (Appendix B) shows the same filtering **consistently improves performance across four model sizes (125M-1.3B) and five benchmarks**. The quality ranking is stable across scales; what changes is the optimal $k$, which shifts with compute budget. A single cheap classifier can be reused without retraining.
>
> We agree that stability across training recipes (e.g., learning rate sensitivity, other architectures) could be studied further, but the recipe used here reflects standard practice.
>
> > **Conditions/failure modes where implicit HQ filtering aligns or misaligns**
>
> There are two main mechanisms. First, alignment succeeds when the HQ set's top CQF-scoring decile emphasizes documents relevant to the target task. Figure 5 shows this: for KnowledgePile, *the top decile exhibits monotonically decreasing loss* as $k$ decreases, and is closest to ARC-Easy in embedding space. Figure 6 validates this at 1.3B scale.
>
> Second, misalignment occurs when:
> -  the HQ set is **too domain-specific** (OpenWebMath causes dips on non-math benchmarks at small k; Figure 2, column 4), or:
> -  the classifier captures **spurious features** (Appendix C, Figure 13: OpenOrca classifier uses sequence length as a quality proxy). We will add a subsection synthesizing these conditions.
>
> > **Detecting/mitigating CQF over-specialization**
>
> Appendix D (Figures 18-20) shows that different HQ sets specialize toward different benchmarks. Figure 20 visualizes this in embedding space and could be used to detect over-specialization.
>
>
>  We will explictely mention two mitigation strategies in the main text:
> -  **using diverse HQ references** (as DCLM combines ELI5 and OpenHermes)
> -  data-conditioning (Section 7) as a diagnostic: if CQF-filtered data fails to be better-conditioned than unfiltered data on a held-out set, the **filtering is too narrow**
>
>
> > **Reproducibility**
>
> We will release the following alongside the revised paper (the ICML code of conduct unfortunately *forbids sharing code* at this stage):
>
> 1. **Pipeline pseudocode**: a self-contained Python file describing the full pipeline end-to-end (CQF data curation, model training, evaluation) using standard libraries (`sklearn`, `sentence_transformers`, `numpy`).
> 2. **Configuration files**:
>    - Model architectures (125M-1.3B) and training hyperparameters matching Table 2
>    - Main CQF experiment (Figures 2, 11): dataset, classifier type, HQ set, k values, benchmarks
>    - CQF vs CRISP comparison (Figure 7): k sweep and cluster settings
>    - 1.3B finetuning setup (Figure 6)
>
> These are sufficient to reproduce every experiment given access to RedPajama-V2 and the HQ sets in Table 1.
>
> We hope these responses address the reviewer's concerns. Again, we thank them for their suggestions which we believe to enhance our paper.

---

> > ### Author Rebuttal · Reviewer_fo8q · 2026-04-01
> >
> > Thank you for your detailed responses to my comments and for addressing the concerns raised. Overall, I am satisfied with the revisions and explanations provided. Your responses have strengthened the paper, and I am happy to raise my score from 4 to 5. Thank you once again for your thoughtful engagement with my feedback.

---

> > > ### Author Response · Authors · 2026-04-02
> > >
> > > Dear reviewer,
> > >
> > > Thank you for your reply, we are happy to read that!

---

### Official Review · Reviewer_2Sgc · 2026-03-07

**Soundness:** 3
**Presentation:** 2
**Significance:** 2
**Originality:** 3
**Overall Recommendation:** 3
**Confidence:** 4

**Summary:**

This paper rethinks the role of classifier-based quality filtering in the LLM pretraining. The authors discover that the model trained with CQF-selected data performs well on downstream benchmarks which having large testing loss on the corresponding high-quality dataset used to construct CQF. Then, the similarity between CQF-selected data with different filtration threshold and the high-quality dataset is analyzed through the lens of PCA projection. The results show that CQF does not select data that resembles the high quality dataset directly but in a way of reweighting and denoising. Furthermore, the authors discussed that CQF may not truly represent the most efficient selection method as it can not perform well under data-conditioning.

**Compliance With Llm Reviewing Policy:**

Affirmed.

**Final Justification:**

This paper investigates the role of CQF in data selection. Overall, the paper is well-presented and the idea is novel. My comments are addressed during the rebuttal stage. However, I am not sure whether the lack of the evaluation of larger models will become the issue of validating the results.

**Key Questions For Authors:**

1. Whether the type of dataset would affect the results
2. Can the perspective of CQF provide some insight on the data selection?
3. Some kinds of CQF are constructed by mixed signals or directly learned by downstream tasks. Can the analysis in this work generalize to thses methods?
4. Can the analysis generalize to larger models (such as 8B?)?

**Limitations:**

I suggest the authors to improve the work by addressing the weakness and the questions.

**Strengths And Weaknesses:**

Strength:
1. The paper investigates the role of CQF and provides a novel perspective of its performance gain.
2. Extensive experiments are conducted to support the arguments, and illustrations are appropriate, making the paper sound.
3. The logic flow is clear.

Waekness:
1. Some of the HQ datasets used in this work are not prepared for general pretraining (e.g., OpenOrca is an instruction dataset mainly for finetuning, OpenWebMath is a dataset focus on math domain), which raises the concern of the inaccurate results.
2. While the perspective of CQF is novel, the paper is mainly about analytical results, and there is no guildline for the data selection based on the analysis, which greatly reduces the contribution. As the filtering process is conducted in terms of expectation (i.e., the loss that classifier is trying to minimize is based on expectation over HQ dataset), the classifier itself cannot imitate the whole distribution of the HQ dataset and can only approach the center of the overall characteristics, which makes the denoising straight forward. Therefore, the value of the finding is limited.
3. Although the general logic flow is clear, some details still make the paper hard to follow. In the beginning of the paper (e.g., abstract, introduction), the authors mentioned that CQF distinguishes data from pretraining data and HQ data, which is confusing as the high-quality data is often regarded as the selected data. I suggest rephrase the wording to make the definition of CQF more clear (e.g., pretraining data -> candidate data, HQ data -> reference data).
4. The experiments are conducted under small models, and whether the results can scale to larger models are unknown.
5. The section of data-conditioning does not greatly fit in the work. The concept of data-conditioning itself is abstract. If the objective of data-conditioning is to evaluate the training speed, it could be substitude with loss curve or the convergence time.

---

> ### Author Rebuttal · Authors · 2026-03-30
>
> We thank the reviewer for acknowledging the novel perspective (S1), extensive experiments (S2), and clear logic (S3). We address each point below, and respectfully note that several concerns rest on misreadings of content already in the paper.
>
> > **Some HQ datasets are not prepared for general pretraining (e.g., OpenOrca, OpenWebMath)**
>
> In CQF, the HQ set is a *reference signal* for the classifier, **not pretraining data**. The model trains on filtered RedPajama-V2, not on the HQ set (cf. Fig. 1, Section 2, lines 127-157). Using diverse HQ references is standard: DCLM (Li et al., 2024) uses ELI5 and OpenHermes as references. Our analysis of how different references produce different biases (Appendix D) is a contribution.
>
> > **there is no guideline for the data selection based on the analysis**
>
> The paper has an **explicit Recommendations section** highlighting that quality filtering primarily removes bad data at scale rather than selecting a "golden" subset, and that small HQ sets are more useful as filter references than for pretraining due to limited diversity. Data-conditioning (Section 7) is proposed as a practical tool for evaluating dataset quality via small proxy models (Section 7.3). We will make these takeaways more prominent.
>
> > **the classifier can only approach the center of the overall characteristics**
>
> We respectfully disagree. This mistakes classification for centroid estimation. **A binary classifier does not estimate the "center" of the HQ distribution**. The Bayes-optimal CQF classifier scores by the likelihood ratio $p_{HQ}(x)/p_{LQ}(x)$ (Section 4, lines 209-218), ranking by distinguishability from the LQ set, not proximity to any mean. Our key finding (Figure 4, Figure 5) is the opposite: CQF upweights HQ examples *far from* the LQ distribution. The implicit filtering phenomenon (Figure 5), the reweighting decomposition (Eq. 1), and the comparison with importance sampling (Section 5, Figure 7) go well beyond "denoising."
>
> > **CQF distinguishes data from pretraining data and HQ data, which is confusing**
>
> We appreciate this and **will clarify terminology** following the conventions in Fig.1.: HQ data = small reference set, pretraining data = large uncurated set, CQF data = selected output.
>
> > **Experiments are conducted under small models**
>
> The paper includes scaling to 1.3B parameters (Figures 6, 10, 11; Section 7.3). **CQF improving models even at large scales is well established in the literature** with >30B models trained (Li et al. 2024, Mizrahi et al. 2025, NAVER Cloud team 2026).
>
> > **data-conditioning could be substituted with loss curve or convergence time**
>
> Data-conditioning measures *cross-dataset transfer* (Eq. 3): $D_{clean}$ is better data-conditioned than $D_{dirty}$ if training on $D_{clean}$ yields lower loss *on $D_{dirty}$* than training on $D_{dirty}$ directly. Loss curves measure within-dataset convergence, they cannot capture that training on one dataset improves performance on a different, noisier dataset. This distinction is made concrete with practical examples in Sections 7.2 and 7.3.
>
> > **Whether the type of dataset would affect the results**
>
> We study this systematically. **Different HQ sets lead to different filtering behaviors and downstream strengths** (Appendix D, Figure 18), which is why understanding the mechanism (Sections 3-5) matters for practitioners.
>
> > **Can the perspective of CQF provide some insight on the data selection?**
>
> The implicit filtering insight (Section 4.2) informs practice: **practitioners should choose HQ sets whose top-scoring deciles align with target tasks** (Figures 5, 6). See also our Recommendations section.
>
> > **Can the analysis generalize to CQF constructed by mixed signals or downstream tasks?**
>
> Our likelihood ratio analysis **applies to any binary classifier**. For mixed-signal CQF, the effective p_HQ becomes a mixture but the core mechanism holds. We **already include an experiment with a downstream-learned classifier** (Fig. 2, rightmost column).
>
> > **Can the analysis generalize to larger models (such as 8B)?**
>
> See W4 above. Work in the literature uses CQF at scales up to 30B+. Our scaling analysis (Section 7.3, Figure 10) provides evidence the findings about data conditioning extrapolate, though we cannot verify at 8B due to compute constraints.
>
> We hope our response clarifies the paper. In light of our answers and the other reviewers' positive assessment, we hope the reviewer will revisit the manuscript accordingly.
>
> Ref: Team, NAVER Cloud HyperCLOVA X. "HyperCLOVA X 32B Think." arXiv:2601.03286 (2026).

---

> > ### Author Rebuttal · Reviewer_2Sgc · 2026-04-01
> >
> > Thanks for answering my comments, I will raise my score.

---

> > > ### Author Response · Authors · 2026-04-02
> > >
> > > Dear reviewer,
> > >
> > > Thank you for increasing your score. We are happy to see that your concerns have been resolved.

---

### Official Review · Reviewer_rccG · 2026-03-13

**Soundness:** 2
**Presentation:** 2
**Significance:** 2
**Originality:** 2
**Overall Recommendation:** 4
**Confidence:** 4

**Summary:**

This paper gives an inner working analysis of the method Classifier-based Quality Filtering (CQF) used to filter high-quality from within a low-quality set. Although CQF is used to filter pretraining data with respect to a high-quality reference set, the authors observe that increasing filtered samples (keeping only higher-quality data) does not improve loss on the reference set itself. They also introduce an optimization-driven new notion of data quality.

**Compliance With Llm Reviewing Policy:**

Affirmed.

**Final Justification:**

The rebuttal was appreciated and addressed my concerns -- as technically solid work that advances at least one sub-area of AI, with a contribution that others are likely to build on, I believe this paper merits a score of 4.

**Key Questions For Authors:**

Please treat weaknesses as questions.

**Limitations:**

Yes.

**Strengths And Weaknesses:**

**Strengths**:
- The paper gives an indepth analyses of the property of high-quality data in terms of CQF.
- This paper introduces a new notion of high-quality data based on learnability.

___

**Weaknesses:**
- Although the paper mentions the architecture and training hyperparameters (line 897), it does not specify the training objective, for example, whether the model is trained using next-token prediction or another loss (e.g. masked token loss).
- It is unclear what intervals of k% are used when reporting performance in Figure 2.
- The study reports performance in Figure 2 using only the ARC-Easy dataset (line 192). It would be beneficial to include results on additional evaluation datasets to rule out the possibility that the observed effect is specific to this task. If contradictory or dissimilar trends appear on other datasets, discussing these differences would help determine whether the observed behavior reflects a more general phenomenon or is specific to ARC-Easy, and whether such differences can still be explained by the current intuition.
- The main text claims that implicit filtering of the HQ set via CQF is beneficial (right column, lines 250-251). However, Appendix D notes that no single HQ set consistently outperforms others across all downstream tasks. This suggests that the benefit of CQF filtering is task-dependent, and the claim in the main text may overgeneralize the effect.
- While the paper claims that training on CQF-filtered data can outperform direct training on the HQ set (right column, lines 300-304), it is unclear whether the HQ set used in the comparison (Figure 8) is the original HQ set or a filtered version. For a fair evaluation of CQF, both the HQ and filtered datasets should contain the same number of examples. Otherwise, performance differences may simply reflect dataset size rather than the effect of quality filtering.
- Although the notion of optimization-driven data quality (left column, lines 32-33) is innovative, it is unclear how the notion identifies examples that are hard to optimize (during training) but important to learn the task.

---

> ### Author Rebuttal · Authors · 2026-03-30
>
> We thank the reviewer for engaging with our work. We are glad the reviewer appreciates the depth of our analysis and the novelty of data conditioning. We address each weakness below, and believe that some concerns rest on misreadings of the paper's content. We hope this response clarifies it.
>
> > **training objective**
>
> We will clarify that we train the models using next-token prediction, which is the most common approach.
>
> > **what intervals of k% are used when reporting performance**
>
> We use k in {1%, 2%, 5%, 10%, 20%, 25%, 50%, 100%}. We will add this explicitly to the Figure 2 caption.
>
> > **The study reports performance in Figure 2 using only the ARC-Easy dataset**
>
> We focus on ARC-Easy because, as explained in the text,  it *"has better-than-random performance even at small scale.”*
>
> Crucially, the same trends are validated on **five downstream benchmarks** (MMLU, MMLC, reward-bench, ARC-Challenge, ARC-Easy) across **four model scales** (125M, 350M, 700M, 1.3B) in **Appendix B, Figure 11** and on 350M models in **Appendix D, Figure 18**. We encourage the reviewer to examine these figures. The  finding  that CQF improves downstream accuracy without necessarily reducing HQ set loss holds across all benchmarks and scales tested, although the trend is clearest for ARC-easy.
>
> To improve clarity, we will add a multi-benchmark version of the Figure 2 top row to the main text in the revision.
>
> > **The main text claims that implicit filtering of the HQ set via CQF is beneficial (right column, lines 250-251). However, Appendix D notes that no single HQ set consistently outperforms others across all downstream tasks. This suggests that the benefit of CQF filtering is task-dependent, and the claim in the main text may overgeneralize the effect.**
>
>
> Thank you for raising this point that we will clarify in the text. These two findings are **complementary, not contradictory**. They address different axes of variation:
>
> - **Section 4.2 (lines 250-251)** concerns implicit filtering *within* a given HQ set: CQF upweights certain HQ examples over others, and this reweighting aligns with downstream tasks (demonstrated in Figures 5 and 6, where the top CQF deciles of the HQ set are shown to be closer to downstream benchmarks).
> - **Appendix D** concerns variation *across* HQ sets: different HQ of references are better suited for different tasks.
>
> We believe there is no contradiction here. Within any chosen HQ set, implicit filtering helps over generic pretraining (e.g. using $k=100$%); across HQ sets, the best choice depends on the target task. Appendix D's title *"No HQ set is superior to all others across all tasks"* explicitly frames this as a property of the CQF landscape, not a limitation of implicit filtering.
>
> > **For a fair evaluation of CQF, both the HQ and filtered datasets should contain the same number of examples.**
>
> We want to highlight that in that experiment, both datasets $D_{HQ}$ and $D_{HQ}'$ **have enough examples** to train the models without ever repeating data, as written explicitely in the text: *"we always use values of k such that there are enough data in $D_{CQF}$ to train a model without repeating data"*.
>
> Therefore, the difference in accuracy cannot be attributed to data scarcity: both models have seen the exact same number of *unique* training examples. We will make this important point more present in the text although it appears already twice, once in bold l.158, right column and l.312, right column.
>
> Studying what happens for  smaller $k$, where a tradeoff between quality and data  scarcity appears, is an interesting avenue for future works.
>
> > **it is unclear how the notion identifies examples that are hard to optimize (during training) but important to learn the task.**
>
> Data-conditioning (Section 7.1, Equation 3) is a **dataset-level** property, not an example-level one. It does not aim to identify individual hard examples. The definition states: $D_{clean}$ is better data-conditioned than $D_{dirty}$ if a model trained on  $D_{clean}$ achieves lower loss on  $D_{dirty}$ than a model trained on $D_{dirty}$ itself. This captures an optimization phenomenon: certain dataset compositions are easier to learn from and transfer better. This arises from the fact that stochastic optimization on noisy data is imperfect (lines 377-384). The concept is analogous to *"conditioning"* in numerical optimization, where a well-conditioned problem is easier to solve, hence the name. This is a new perspective on quality that we introduce which is tangential to the one you mentioned.
>
> We demonstrate this property empirically in Figure 9 and validate its stability across model scales (125M to 1.3B) using Chinchilla-style scaling laws in Section 7.3 and Figure 10. We will expand the exposition to make the dataset-level nature of the definition clearer.
>
>
> We thank you again for reviewing our paper, and hope that we have addressed your concerns accordingly.

---

> > ### Author Rebuttal · Reviewer_rccG · 2026-04-01
> >
> > Thank you for addressing my concerns. I have increased my rating.

---

> > > ### Author Response · Authors · 2026-04-02
> > >
> > > Dear reviewer,
> > >
> > > Thank you for your reply, we are happy to read that!

---

### Official Review · Reviewer_dDEQ · 2026-03-13

**Soundness:** 3
**Presentation:** 3
**Significance:** 3
**Originality:** 2
**Overall Recommendation:** 5
**Confidence:** 4

**Summary:**

The paper investigates the effects of using binary discrimination filters (or classifiers) for performing dataset curation. These techniques have become very common in the community and the paper questions whether their purpose is about "identifying examples similar to the high-quality data" or "excluding low-quality data". Through a series of extensive experiments, they show that these discrimination-based classifiers do not improve the loss on the high-quality subset, but still help with inducing capabilities. Finally, the authors introduce the lens of "data conditioning" which tries to formalize "low-quality data" from an optimization and loss perspective.

**Compliance With Llm Reviewing Policy:**

Affirmed.

**Final Justification:**

My positive opinion from my original review stands. However, I think it's important to rephrase some of the claims around novelty in the paper (which was more or less acknowledged in the rebuttal)

**Key Questions For Authors:**

> In this paper, we step away from this limitation and always use values of k such that there are enough data in DCQF to train a model without repeating data.

Is this simply achieved by ensuring there is access to a large enough "source corpus"?

Was the training carefully deduplicated?

**Limitations:**

yes

**Strengths And Weaknesses:**

Strengths:

* I appreciate the precise and relevant focus of the paper on the effects of classifier-based data selection. The insights are backed up with good empirical evidence and they appear genuinely useful to me for the language modeling and the wider machine learning community.
* Despite operating at small scales, the authors run many experiments by smoothly varying the data selection strategy to understand trends. I think the experimental settings and target benchmarks are well-chosen for the purpose of the ppaer. I also find the small-scale scaling experiments in section 7.3 very interesting and these should be a good starting point for future work.
* The framework of data-conditioning in this context is interesting and a novel perspective for expressing qualities of low-quality data in my view.

Weaknesses:
* Some insights of the paper that are presented as novel contributions are already covered by prior work. Therefore, the main thesis of section 5 "CQF is not importance sampling" should come as no surprise to researchers in this area. Instead, the authors here may refer to "Data Selection for Language Models via Importance Resampling" (DSIR) by Xie et al., NeurIPS 2023 which makes this observation and introduces an improved method for matching target distributions. I also believe "CoLoR-Filter: Conditional Loss Reduction Filtering for Targeted Language Model Pre-training" by Brandfonbrener et al., NeurIPS 2025 is highly relevant in this context. It may be worth discussing that GPT-3 data selection employed a stochastic selection criterion based on samples from a pareto distribution. Nevertheless, I think the paper still makes a useful contribution here since simple threshold-based selection based on classifier scores *has* remained the most popular technique for large-scale data selection (over DSIR and stochastic thresholding).
* Claims about the precision of quality classifiers should also take into account the capacity and inductive bias of the selection classifier (e.g. linear predictor based on embedding features vs fasttext vs small neural models), and this aspect is somewhat overlooked in the otherwise comprehensive paper.
* Some aspects of the language and methodology in the paper, e.g. "Kullback-Leibler divergence between datasets" could be made more precise

---

> ### Author Rebuttal · Authors · 2026-03-30
>
> We thank the reviewer for the careful and constructive review, and for recognizing the relevance of our work to the ML community. We appreciate the specific and actionable suggestions, and address each weakness/question below.
>
> > **Relation to prior work** (DSIR, CoLoR-Filter, GPT-3 selection)
>
> We agree that DSIR (Xie et al., 2023) and CoLoR-Filter (Brandfonbrener et al., 2025) are relevant and should be discussed more prominently. Thank you for suggesting it. We will add a dedicated discussion in the revised manuscript. Additionally, we want to clarify the distinct contributions of our work relative to these methods:
>
> - **DSIR** proposes an alternative to CQF by performing importance resampling to match the HQ distribution. While the main idea behind DSIR is indeed that standard CQF does not perform importance sampling, their contribution is a new method, not an analysis of why threshold-based CQF still works well in practice. Our paper fills this gap: we show that CQF's effectiveness comes from noise removal and implicit HQ filtering (Section 4.2), not from distribution matching. In fact, our comparison with importance sampling (we use CRISP, which in spirit is very close to DSIR)  in Section 5 (Figure 7) shows that CQF and importance sampling produce qualitatively different outcomes: CQF induces a filtering of the HQ set that importance sampling does not, which explains CQF's superior downstream performance when the HQ set is not the downstream task itself.
>
> - **CoLoR-Filter** scores data by the loss reduction between two auxiliary LMs: a marginal model trained on prior data and a conditional model fine-tuned on downstream task data. It selects data points where the conditional model assigns higher likelihood than the marginal model. This is explicitly task-targeted, it requires access to downstream task data at selection time. CQF uses a generic HQ reference without knowledge of the downstream task. Our analysis of CQF's implicit mechanisms (Sections 4, 5) is specific to classifier-based filtering and does not directly apply to loss-reduction methods like CoLoR-Filter. However, our data-conditioning framework provides an interesting avenue for comparing both methods: we could evaluate whether CoLoR-Filter-selected data is better data-conditioned than CQF-selected data. We will discuss this connection in the revision.
>
> - **GPT-3's  stochastic selection**: good point, we will extend the discussion around stochastic thresholding on page 2. It acts as a "soft-CQF" which favors high likelihood ratios but without a hard cutoff. We will extend the analysis in Sec.4 to also cover that case.
>
> We appreciate the reviewer's acknowledgement that our contribution remains valuable given that simple threshold-based CQF is still the dominant technique.
>
> > **Classifier capacity and inductive bias**
>
> Indeed, our experiments use a single classifier architecture (L2-regularized logistic regression on text embeddings, as used by most practitioners). While we **investigate classifier biases** in Appendix C where we show that CQF classifiers can learn spurious features such as sequence length (Figure 13) and that debiasing strategies can mitigate this (Figure 14), we do not systematically vary the classifier architecture itself; this is an interesting future work direction.
>
> > **Precision of language** (e.g., "KL divergence between datasets")
>
> We acknowledge this point and **will tighten the language** in the revision. By *KL divergence between datasets* we mean the KL divergence between the probability distributions that generated those datasets, $\mathrm{KL}(p_{\mathrm{HQ}} || p_{\mathrm{CQF}})$. This is approximated with models that have sufficient capacity to represent the data-generating distribution (as we state in the assumption on l. 258). We will make this sharper throughout the manuscript. Thank you.
>
> > **Is [having enough CQF data] achieved by ensuring access to a large enough source corpus?**
>
> Indeed. Our LQ source corpus is RedPajama-V2, which contains 32T tokens (Section 2, l. 141). Even at aggressive filtering ($k = 1$%), this yields approximately 320B tokens, which is sufficient to train our largest models without any repetition. Studying what happens for even smaller k, where a tradeoff between quality and data repetition appears, is an interesting avenue for future works.
>
> > **Was the training carefully deduplicated?**
>
> RedPajama-V2 applies document-level deduplication as part of its standard pipeline (Weber et al., 2024). We did not apply additional deduplication beyond what is provided by RedPajama-V2, as our focus is on studying the effect of quality filtering rather than deduplication. We will clarify this in the revision. Studying the interaction between deduplication and CQF is another interesting direction but orthogonal to the core contributions of this paper.
>
> Thank you again for the constructive suggestions that help us improve the paper !

---

> > ### Author Rebuttal · Reviewer_dDEQ · 2026-04-05
> >
> > I think it remains important to weaken some of the claims around novelty in the introduction and some of the sections, e.g. regarding importance sampling, rather than simply citing these works in the related works.
> >
> > Otherwise, I like the paper!

---

### Decision · Program_Chairs · 2026-04-30

**Decision:**

Accept (regular)

**Comment:**

This paper provides a critical analysis of Classifier-based Quality Filtering (CQF), a ubiquitous method for curating large-scale web data for language model pretraining. The authors present the counterintuitive finding that CQF improves downstream task performance without necessarily reducing loss on the "gold" reference dataset, and that training on CQF-filtered data can outperform training directly on the (sufficiently large) high-quality reference set itself. The paper attributes this to an implicit filtering/reweighting effect within the high-quality set and introduces an optimization-driven concept called "data-conditioning" to reframe how we evaluate data quality.

This paper finally received four reviews with scores 1x Weak reject, 1x Weak accept, 2x Accept. It is noteworthy that the reviewer who gave the recommendation of Weak reject marked all concerns as Fully Resolved in the rebuttal acknowledgement and stated, "I will raise my score". However, the final numeric score in the system remains 3 (Weak Reject). So I consider that all the reviewers have reached an agreement on the acceptance of this paper. I recommend that the authors incorporate the promised revisions regarding taxonomy (as suggested by Reviewer fo8q) and the adjustment of novelty claims (as suggested by Reviewer dDEQ) into the final manuscript.

I recommend Accept.